# BENCHMARKING AND RETHINKING MULTIPLEX GRAPHS

## ABSTRACT

Multiplex graphs, which represent complex real-world relationships, have recently garnered significant research interest. However, contemporary methods exhibit variations in implementations and settings, lacking a unified benchmark for fair comparison. Additionally, existing multiplex graph datasets suffer from small-scale issues and a lack of representative features. Furthermore, current evaluation metrics are restricted to node classification and clustering tasks, lacking evaluations on edge-level tasks. These obstacles impede the further development of the multiplex graph learning community. To address these issues, we first conducted a fair comparison based on existing settings, finding that current methods are approaching performance saturation on existing datasets with minimal differences; and simple end-to-end models sometimes achieve better results. Subsequently, we proposed a unified multiplex graph benchmark called MGB. MGB includes ten baseline models with unified implementations, formalizes seven existing datasets, introduces four new datasets with text attributes, and proposes two novel edge-level evaluation tasks. Experiments on MGB revealed that the performance of existing methods significantly diminishes on new challenging datasets and tasks. Additional results suggest that models with global attention and stronger expressive power in end-to-end solutions hold promise for future work. The data, code, and documentations are publicly available at `https://anonymous.4open.science/r/multiplex-F150`.

## 1 INTRODUCTION

In recent years, the field of graph learning has witnessed rapid development (Kipf & Welling, 2016; Veličković et al., 2017). Multiplex graphs (Zhang et al., 2018), which incorporate diverse relationships between nodes, offer a more realistic representation of multiple structural connections between nodes in the real world, attracting considerable research interest. An example of a multiplex graph is an e-commerce network (Ni et al., 2019), where different products have multiple types of relationships, including co-purchased, co-viewed, and complementary connections. Different connection types play distinct roles in specific contexts. It's noteworthy that in real-world applications, multiplex graphs also come with rich textual attributes, such as product descriptions in e-commerce networks and paper abstracts in citation networks. Data with such *structural multiplicity* and *attributive richness* holds great potential in applications like knowledge graph construction (Zhao et al., 2022), recommendation systems (Zhang et al., 2020), and anomaly detection (Guo et al., 2024), among others.

Despite the rapid development of multiplex graph learning, three key issues persist in the field: **(i) Inconsistent Comparisons:** Different methods adopt unique data processing, model implementation, and experimental settings, hampering our ability to comprehensively understand them and making fair comparisons challenging. **(ii) Insufficient Datasets:** Common multiplex graph datasets often have a limited scale, containing only a few thousand nodes. Furthermore, despite containing rich raw textual information, the original data is typically encoded into vectors using shallow embedding methods, which restricts expressiveness and generalization. Consequently, existing multiplex graph works primarily focus on representation learning, especially self-supervised representation learning, employing complicated data augmentation and contrastive paradigms to learn features, which is time-consuming and resource-intensive. **(iii) Limited Evaluation Metrics:** Existing methods incorrectly train and evaluate the inherently multi-labeled IMDB (Wang et al., 2019b) dataset using a single-label strategy. Additionally, these methods primarily evaluate models on node-level tasks, neglecting

edge-level tasks with multiplex relationships. This limitation restricts our understanding of the models' ability to learn structural information beyond node attributes.

To address the challenge of inconsistent comparisons, we first reproduced ten classic methods within a unified framework, evaluating these methods based on a consistent set of hyperparameters and dataset processing pipelines. Building upon this, we constructed the Multiplex Graph Benchmark (MGB) to further address the challenges of insufficient datasets and limited evaluation metrics. Currently, MGB includes *10 state-of-the-art methods* (with unified interfaces for model implementation and training), *11 multiplex datasets* (comprising 7 commonly used and 4 newly curated datasets with raw text attributes), and *4 evaluation tasks* (including node classification and clustering, edge prediction and classification). Figure 1 shows a comparison of the scale of datasets in MGB with other common datasets.

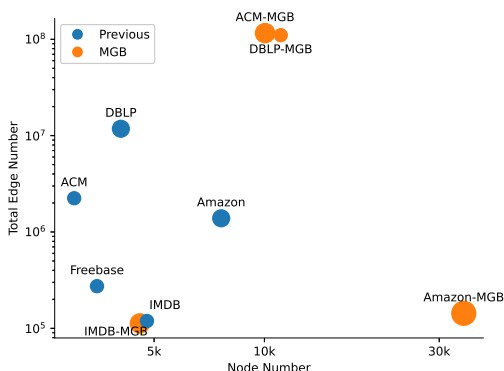

Figure 1: Comparison of dataset scales. The node size corresponds to the number of classes of each dataset.

By comparing methods under existing setting and MGB setting (refer to Table 1, 4 respectively), we observed several key phenomena: Firstly, existing methods have almost approached performance saturation on five previous small-scale datasets, making it challenging to compare and evaluate different approaches. Secondly, end-to-end methods, even though implemented using simple GNNs, achieve remarkable performance compared to self-supervised methods. Thirdly, existing methods exhibit poor performance on four new datasets. These new datasets are more sparse, rich in feature space, and label space, making them more challenging and requiring models with larger capacity and expressivity. Lastly, we explored a possible research direction through a simple Graph Transformer model, suggesting that end-to-end models that better capture global relationships and understand deep features could be more effective.

In conclusion, our contributions can be summarized as follows:

- **Reproducible and Fair Comparison:** We are the first, to our best knowledge, to conduct a fair comparison in the multiplex graph field by standardizing the implementation of different methods, setting hyperparameters uniformly, and using datasets with the same versions and configurations. Our experimental results highlight the limitations of current small-scale datasets and simple tasks in the field of multiplex graph learning.

- **New Challenging Benchmark:** Building on the aforementioned fair comparison, we introduce MGB, a unified benchmark for multiplex graphs, including scalable baseline implementations, larger and more challenging datasets with text attributes, and novel edge-level tasks to propel multiplex graph research forward.

- **Empirical Findings:** By comparing baseline methods on MGB, we find that existing methods perform significantly worse on the MGB datasets, with a notable increase in the performance gap between models. We also highlight opportunities and possible directions for future work, suggesting the need for deeper and more robust models.

## 2 PRELIMINARIES

### 2.1 TASK FORMULATION

**Multiplex Graphs**  A multiplex graph is a network consisting of $\mathcal{G} = \{\mathcal{G}_1, \mathcal{G}_2, ..., \mathcal{G}_R\}$, where $\mathcal{G}_r = \{\mathcal{V}, \mathcal{E}_r, \mathbf{A}_r, \mathbf{X}\}$ is the $r$-th subgraph of the multiplex graph corresponding to the $r$-th meta-path (also known as relationship or view), and $R$ denotes the number of subgraphs. For each $\mathcal{G}_r$, $\mathcal{V}$ and $\mathcal{E}_r$ denote the node set and edge set, respectively; $\mathbf{A}_r \in \mathbb{R}^{|\mathcal{V}| \times |\mathcal{V}|}$, and $\mathbf{X} \in \mathbb{R}^{|\mathcal{V}| \times d}$ represent the adjacency matrix and feature matrix. It is important to note that all $\mathcal{G}_r$ share the same node set $\mathcal{V}$ and feature matrix $\mathbf{X}$ but have different edge sets $\mathcal{E}_r$ and adjacency matrices $\mathbf{A}_r$.

**Multiplexity vs. Heterogeneity** Multiplex graphs and heterogeneous graphs (Lv et al., 2021; Zhang et al., 2019) are two distinct subsets of multi-relational graphs (Hamilton). Heterogeneous graphs feature diverse types of nodes and edges, with edges typically constrained based on node types, often connecting nodes of specific types. Conversely, multiplex graph focuses on multiple interactions between the same pairs of nodes (Melton & Krishnan, 2023; Yu et al., 2022). In essence, heterogeneous graphs emphasize connections between specific node types, while multiplex graphs prioritize interactions across different relation types. Therefore, the research methods and focuses of these two areas differ to some extent.

## 2.2 Methods on Multiplex Graph

The majority of research in the field of multiplex graph learning focuses on representation learning, which involves learning node embeddings by integrating information from multiple meta-paths. In this paper, we roughly classify existing work into two categories based on the training objectives.

**(i) End-to-end methods** take raw node features and graph structure information as model inputs, and the output node embeddings are directly used for specific downstream tasks such as node classification. HAN (Wang et al., 2019b) aggregates multiple node embeddings using hierarchical attention at different levels and trains the model using cross-entropy loss with ground-truth labels. Traditional Graph Neural Networks (*e.g.*, GCN (Kipf & Welling, 2016), GAT (Veličković et al., 2017)) can also be trained by directly using average-readout on multiple node embeddings. **(ii) Self-supervised methods** do not rely on ground-truth labels. Instead, they mostly utilize contrastive learning (Chen et al., 2020; Grill et al., 2020; He et al., 2020) to help the model learn general representations. MNE (Zhang et al., 2018) and DMG (Mo et al., 2023b) learn a common embedding and a private embedding for each subgraph, then combine them using attention mechanism. DMGI (Park et al., 2020), HDMI (Jing et al., 2021), and SSDCM (Mitra et al., 2021) leverage contrastive learning by maximizing mutual information between node embeddings and graph readouts. HeCo (Wang et al., 2021), CKD (Wang et al., 2022a), and MGDCR (Mo et al., 2023a) contrast between different subgraphs and within individual subgraph. Other works also focus on learning scalable embeddings (Liu et al., 2020), dealing with incomplete data (Wang et al., 2022b), and so on. A common issue among the self-supervised methods is the requirement for complicated contrastive paradigms and resource-consuming negative sampling.

In addition, multiplex graph learning also demonstrates the ability to effectively characterize complex relationships in real-world applications. For example, CS-MLGCN (Behrouz & Hashemi, 2022) explores the application of multiplex graphs in community search. ANOMULY (Behrouz & Seltzer, 2022) combines multiplex graphs with dynamic graphs for anomaly detection tasks on time-series multiplex graph data, including scenarios such as blockchain security and brain disease prediction. ADMire (Behrouz & Seltzer, 2023) also utilizes multiplex graphs to model brain networks and detect anomalies in the human brain.

Appendix B provides more detailed information about each mentioned work. However, due to the different settings and implementations of each method, achieving a unified fair comparison is challenging, impeding further development in multiplex graph learning.

## 2.3 Existing Datasets and Evaluations

**Datasets** We consider seven commonly used multiplex graph datasets, including two citation networks: ACM (Wang et al., 2019b) and DBLP (Gao et al., 2009), two movie review networks: IMDB [1] and Freebase (Yang et al., 2020), one commercial network: Amazon (Ni et al., 2019), and two anomaly detection networks: Amazon-fraud and Yelp-fraud (Dou et al., 2020). These datasets vary in size, with node numbers ranging from 3k to 5k (except for the two anomaly detection datasets). Each dataset contains 2 or 3 multiplex meta-paths. Due to the existence of multiple versions (Fu et al., 2020; Lv et al., 2021; Wang et al., 2019a) of these datasets, we adopt the most widely used versions to ensure fair comparison, namely ACM, IMDB, and DBLP from HAN (Wang et al., 2019b), Amazon from DMGI (Park et al., 2020), and Freebase from HeCo (Wang et al., 2021). It is worth noting that the authors of (Park et al., 2020; Wang et al., 2019b; 2021) are not the original collectors

---

[1]https://www.kaggle.com/datasets/karrrimba/movie-metadatacsv

and annotators of the raw datasets, and the credit for raw data collection can be found in the Appendix A. The detailed statistics of each dataset are provided in Table 3.

Existing multiplex graph datasets are small in scale, and their annotators have provided oversimplified embeddings (e.g., bag-of-words or one-hot encoding) as initial node features. These limitations may introduce potential issues, which will be discussed in the fair comparison in the next section.

**Evaluations**   Existing studies utilize two node-level tasks to evaluate multiplex graph methods: node classification and node clustering. For the node classification task, Macro-F1 and Micro-F1 metrics are utilized to evaluate models on validation and test sets. For the node clustering task, common metrics including Accuracy, F1 score, Normalized Mutual Information (NMI), and Adjusted Rand Index (ARI) are used to assess performance. Moreover, DMGI (Park et al., 2020) and HDMI (Jing et al., 2021) introduce top-K similarity search (Sim@K) as an additional metric for evaluating clustering performance. First, they compute the cosine similarity between each pair of nodes. Then, for each node, calculate the proportion of nodes with the same label among its top-K similar nodes. Typically, K is set to 5.

# 3   FAIR COMPARISON UNDER EXISTING SETTING

Table 1: Fair comparison results under existing settings, where higher value is better. Colored are the best first, second, and third results. *E2E* denotes end-to-end methods, *SS* denotes self-supervised methods. *As HeCo (Wang et al., 2021) is fundamentally a heterogeneous graph method requiring different types of nodes, we only reproduced its results on DBLP and Freebase.

(a) Node classification results reported in Macro-F1 and Micro-F1.

|  | Methods | ACM | | IMDB | | DBLP | | Amazon | | Freebase | |
|---|---|---|---|---|---|---|---|---|---|---|---|
|  |  | Macro-F1 | Micro-F1 | Macro-F1 | Micro-F1 | Macro-F1 | Micro-F1 | Macro-F1 | Micro-F1 | Macro-F1 | Micro-F1 |
| E2E | GCN | 81.83±1.3 | 82.35±1.1 | 57.02±0.2 | 58.41±0.2 | 90.60±0.2 | 91.47±0.1 | 63.01±0.3 | 63.42±0.2 | 52.66±2.4 | 54.92±2.7 |
|  | GAT | 67.58±19.5 | 72.63±13.7 | 55.44±0.6 | 57.60±0.8 | 87.60±1.9 | 88.65±2.1 | 52.95±0.7 | 54.51±0.6 | 52.12±1.6 | 53.26±2.0 |
|  | HAN | 83.81±0.6 | 83.75±0.6 | 54.89±0.6 | 57.25±0.5 | 87.81±3.0 | 88.92±3.1 | 59.30±1.2 | 60.08±1.0 | 50.93±0.8 | 51.88±0.9 |
| SS | MNE | 73.74±0.8 | 74.80±0.7 | 45.69±1.0 | 46.21±1.0 | 71.43±0.3 | 72.50±0.3 | 69.50±0.8 | 69.95±0.7 | 32.64±0.8 | 33.94±0.9 |
|  | DMGI | 86.08±0.8 | 86.08±0.8 | 46.91±3.2 | 48.22±3.6 | 87.18±0.9 | 88.34±0.8 | 66.52±1.7 | 67.25±1.8 | 37.24±0.8 | 38.40±1.1 |
|  | HDMI | 89.58±0.6 | 89.56±0.6 | 50.33±1.8 | 50.93±1.9 | 92.53±0.3 | 93.39±0.2 | 62.77±2.9 | 63.43±2.9 | 50.86±1.7 | 52.60±2.0 |
|  | HeCo* | NA | NA | NA | NA | 88.01±0.8 | 88.93±0.7 | NA | NA | 37.95±2.2 | 39.10±2.4 |
|  | CKD | 87.90±0.9 | 87.91±0.9 | 49.26±2.4 | 51.10±2.1 | 89.14±0.3 | 90.26±0.4 | 61.06±1.6 | 62.24±1.6 | 47.00±1.9 | 49.76±1.6 |
|  | MGDCR | 80.74±5.2 | 80.82±5.0 | 50.69±5.7 | 52.20±5.5 | 87.62±8.7 | 88.43±8.5 | 63.56±2.7 | 64.14±2.5 | 41.36±7.2 | 43.46±6.9 |
|  | DMG | 88.56±1.3 | 88.56±1.3 | 52.34±2.4 | 55.07±2.0 | 91.65±0.1 | 92.59±0.1 | 43.38±3.9 | 44.36±3.8 | 48.18±3.1 | 49.94±3.5 |

(b) Node clustering results reported in Accuracy and Normalized Mutual Information (NMI).

|  | Methods | ACM | | IMDB | | DBLP | | Amazon | | Freebase | |
|---|---|---|---|---|---|---|---|---|---|---|---|
|  |  | Accuracy | NMI | Accuracy | NMI | Accuracy | NMI | Accuracy | NMI | Accuracy | NMI |
| E2E | GCN | 86.78±0.3 | 62.34±0.5 | 58.42±0.2 | 14.14±0.2 | 90.79±0.2 | 71.70±0.3 | 53.12±0.3 | 20.26±0.2 | 59.54±0.5 | 15.46±0.1 |
|  | GAT | 86.66±0.1 | 61.83±0.2 | 57.72±0.5 | 12.97±0.3 | 89.94±0.3 | 69.96±0.4 | 50.98±0.6 | 14.84±0.3 | 60.13±1.0 | 15.45±0.2 |
|  | HAN | 84.23±0.7 | 58.53±0.9 | 56.57±0.4 | 12.60±0.3 | 87.33±1.3 | 64.35±2.7 | 59.58±0.8 | 20.50±0.5 | 61.64±0.9 | 17.71±0.6 |
| SS | MNE | 51.70±2.1 | 16.07±2.8 | 39.71±0.2 | 2.36±0.2 | 36.75±0.6 | 5.30±1.9 |  |  | 38.65±0.5 | 0.21±0.0 |
|  | DMGI | 85.25±1.3 | 60.42±1.8 | 48.53±2.9 | 8.22±2.1 | 82.07±3.1 | 58.42±5.0 | 58.03±3.2 | 27.93±2.9 | 35.46±0.3 | 0.18±0.0 |
|  | HDMI | 90.77±0.5 | 71.82±0.9 | 53.26±2.8 | 10.19±2.5 | 89.04±1.0 | 71.58±1.2 | 52.72±3.4 | 22.53±3.8 | 54.40±1.7 | 17.33±0.9 |
|  | HeCo* | NA | NA | NA | NA | 68.61±5.8 | 42.63±4.5 | NA | NA | 47.23±2.1 | 2.53±0.7 |
|  | CKD | 86.05±3.8 | 61.91±6.1 | 52.04±1.7 | 9.64±1.4 | 87.02±1.1 | 66.20±0.9 | 36.34±0.6 | 6.05±0.5 | 49.12±2.4 | 9.52±3.8 |
|  | MGDCR | 71.67±9.8 | 53.50±3.6 | 43.89±5.5 | 4.08±2.9 | 80.23±24.7 | 61.37±30.3 | 38.02±3.9 | 10.22±4.3 | 47.94±7.1 | 6.58±6.8 |
|  | DMG | 76.27±2.8 | 56.52±2.3 | 53.89±1.4 | 12.83±1.2 | 87.53±2.8 | 69.98±3.2 | 35.49±2.5 | 6.47±3.6 | 50.00±3.2 | 10.40±3.5 |

The lack of fair comparisons has hindered our ability to reasonably assess the differences between methods and guide future developments. Therefore, we provide a fair comparison based on the common methods, datasets, and evaluation metrics mentioned in Section 2. Specifically, for all compared methods, we consistently set the training epochs to 200, the learning rate within the range of [1e-4, 1e-3, 1e-2], and the weight decay to 1e-4. If a method uses a GNN model, we implement it with a 2-layer GCN encoder and a hidden size in the range of [64, 128]. We report the mean and standard deviations of five runs with random seeds [0, 1, 2, 3, 4]. For our implementations, we use the default configurations from the original papers if provided; otherwise, we employ grid search for hyperparameter tuning. More hyperparameter details about each baseline method are provided in Appendix C.1. Due to space limit, we leave the results and analysis on two anomaly detection datasets (Amazon-fraud and Yelp-fraud (Dou et al., 2020)) in Appendix E.1. The main results are summarized in Table 1, yielding several noteworthy observations:

**Observation 1. Inconsistency with originally reported performance.** Taking CKD (Wang et al., 2022a) as an example, its originally reported Macro-F1 and Micro-F1 scores on the ACM dataset ranges between 91.9 and 92.9. However, DMG (Mo et al., 2023b) reports CKD's results as 90.5, while DMG itself scores 91.0. Neither paper includes implementations of the other methods in their code, making it difficult to judge the relative merits of these methods based solely on the original data. In Table 1a and 1b, we observe that CKD's classification performance on ACM dataset is indeed slightly lower than DMG, but CKD surpasses DMG in clustering performance. Additionally, HDMI (Jing et al., 2021) used an IMDB version with 3,550 nodes in its original paper, which is inconsistent with the versions used by all other methods. Its node classification performance in the original paper often exceeds 60%. However, the results in Table 1 allow us to fairly compare HDMI's performance against a consistent benchmark.

**Observation 2. Minor difference in performance among methods on existing datasets for node classification task.** As shown in Table 1a, the performance values for each method are closely clustered, making it challenging to discern methodological differences. For example, on the DBLP (Wang et al., 2019b) dataset, all methods except MNE (Zhang et al., 2018) fall within the performance range of 87-92, showing very low differentiation. Similarly, on the IMDB dataset, classification results are clustered within the 45-55 range. We speculate that this phenomenon may stem from the modest scale of existing datasets and the limited classification space, rendering the task relatively simple. Consequently, current methods have nearly reached the performance ceiling on these datasets. Additionally, the evaluation metrics are overly simplistic, relying solely on node classification and clustering tasks, which makes it difficult to comprehensively distinguish the performance of different models.

**Observation 3. End-to-end methods achieve comparable or superior results to other self-supervised methods.** Surprisingly, end-to-end methods (i.e., GCN (Kipf & Welling, 2016), GAT (Veličković et al., 2017), HAN (Wang et al., 2019b)) exhibit competitive performance despite their simple implementation without intricate aggregation designs for various meta-paths. Notably, for node classification in Table 1a, end-to-end methods surpass self-supervised methods by a significant margin on the IMDB and Freebase datasets. For node clustering in Table 1b, end-to-end methods also significantly outperform self-supervised methods on the IMDB, DBLP, and Freebase datasets. This observation aligns with the findings reported in Li et al. (2023) and Lv et al. (2021), prompting us to reconsider whether a straightforward end-to-end model may be more suitable given the relatively modest dataset size in the current multiplex graph domain.

## 4 MGB: A Unified Multiplex Graph Benchmark

Based on the results and analysis in Section 3, we summarize some issues existing in the field of multiplex graph learning, which motivate the proposal of MGB.

**Issues with current datasets** First, existing datasets are often small and simplified, lacking the complexity needed to reflect real-world scenarios. For instance, widely used versions of the ACM, IMDB, and DBLP datasets (Wang et al., 2019b) contain only 3,000 to 5,000 nodes, and their classification tasks are limited to mostly three-class problems. This scale is relatively small given the rapid development of current graph datasets. **Another limitation is the absence of high-quality features in current datasets**. The original data contains a wealth of textual information, but previous datasets often discard this information or process it using shallow embedding methods, such as bag-of-words. In some cases, like Freebase (Yang et al., 2020), no features are provided at all. This lack of high-quality features constrains models' abilities to capture complex relationships in the data. These issues are reflected in Observation 2 of Section 3, where various methods approach performance ceilings on existing datasets, reducing their discriminative ability.

**Issues with current evaluations** There are errors in existing classification tasks. Specifically, existing evaluations on the IMDB dataset involve only single-label classifications, whereas the categorizations in the IMDB dataset are actually multi-labeled. For example, the movie "Rush Hour 3" belongs to Action, Comedy, and Thriller categories simultaneously. In such cases, the label space may overlap, making single-label classification unreasonable. **Moreover, existing methods only evaluate models at the node level, lacking evaluation for edge tasks**. In multiplex graphs, different meta-paths represent different types of edges, inherently providing conditions for edge classification

and prediction tasks. However, previous works have not evaluated edge tasks, missing an opportunity to assess models' performance on a crucial aspect of multiplex graph analysis.

### 4.1 BASELINES IMPLEMENTATION

As reiterated in Section 3, we have implemented ten well-recognized multiplex graph learning methods within a unified training and evaluation framework using PyTorch and PyG (Fey & Lenssen, 2019). Among these ten methods, three (GCN (Kipf & Welling, 2016), GAT (Veličković et al., 2017), HAN (Wang et al., 2019b)) are trained end-to-end, while the remaining seven (MNE (Zhang et al., 2018), DMGI (Park et al., 2020), HDMI (Jing et al., 2021), HeCo (Wang et al., 2021), CKD (Wang et al., 2022a), MGDCR (Mo et al., 2023a), DMG (Mo et al., 2023b)) are trained in a self-supervised fashion to learn node embeddings. For these self-supervised methods, a task-specific classifier is subsequently trained on labeled data to evaluate downstream tasks.

A brief introduction to each method is provided in Appendix B. For all reproduced methods, we offer scalable implementations and maintain consistent interfaces to ensure fair comparisons. We will continue to update and include more baseline methods in the future.

### 4.2 DATASETS CONSTRUCTION

**Data Preparation**   We constructed four new datasets for multiplex graph tasks by gathering the raw tabular files of commonly used datasets, including ACM (Wang et al., 2019b), IMDB, DBLP (Gao et al., 2009), and Amazon[2] (Ni et al., 2019). The raw data underwent cleaning and denoising processes. We then expanded the dataset scales, increased category spaces, and augmented the text features based on dataset characteristics. The data was divided into training, validation, and testing sets with ratios of 0.2, 0.1, and 0.7, respectively.

Table 2: Examples of textual attributes of the proposed datasets in MGB.

| Dataset | Text Content | Example |
|---|---|---|
| ACM-MGB | title, abstract, authors, venue | Title & Abstract: Influence and correlation in social networks. In many online social systems, social ties between users play an important role...; Authors: Mohammad Mahdian, Ravi Kumar, Aris Anagnostopoulos; Venue: Proceedings of the 14th ACM SIGKDD international conference on Knowledge discovery and data mining |
| IMDB-MGB | title, director, keywords, plots | Title: Avatar; Director: James Cameron; Keywords: avatar — future — marine — native — paraplegic; Plot: A paraplegic Marine dispatched to the moon Pandora on a unique mission becomes torn between following his orders and protecting the world he feels is his home. |
| DBLP-MGB | title, authors, abstract | Title: Action Recognition with Trajectory-pooled Deep convolutional Descriptors; Authors: Limin Wang, Xiaoou Tang; Abstract: Visual features are of vital importance for action understanding... |
| Amazon-MGB | title, brand, description | Title: OXO Tot Silicone Drying Mat, White; Brand: OXO; Description: Slim + flexible = The ultimate Drying Mat. Efficiently dry baby bottles, sippy cups, breast pump parts, and more with the OXO Tot Silicone Drying Mat. The Drying Mat's rib design maximizes aeration and elevates items, keeping them clean. |

**Adding Textual Features**   The recent surge in large language model (LLM) (Achiam et al., 2023; Anil et al., 2023; Touvron et al., 2023) has significantly enhanced machines' natural language understanding and processing abilities. This development influenced graph learning, giving rise to textual-attributed graphs (He et al., 2023; Yan et al., 2023), merging graphs with LLM and presenting new research opportunities and challenges (Liu et al., 2023; Wang et al., 2024). Consequently, in constructing our new multiplex graph datasets, we introduced rich textual information to support possible future research. For instance, in IMDB-MGB, we extracted textual features of movie titles, directors, actors, and keywords from the original CSV data. Additionally, we scraped movie plots from public websites, forming more comprehensive textual features in the new dataset. We also employed pre-trained language models such as Sentence-BERT (Reimers & Gurevych, 2019) to obtain more expressive features than bag-of-words model. Table 2 shows the detailed information on text features for each dataset.

---

[2]https://cseweb.ucsd.edu/~jmcauley/datasets.html#amazon_reviews

**Expanding Dataset Scales** To accommodate the increasing scale of graph data and models, we expanded the datasets by increasing the number of nodes based on the raw data and adding more meta-paths for multiplex graphs. Specifically, our proposed Amazon-MGB dataset includes seven categories, with 5000 nodes per category, totaling 35,000 nodes, making it one of the largest datasets in the current multiplex graph domain. Additionally, our ACM-MGB and DBLP-MGB datasets have been scaled up to $2.7\times$ to $3.3\times$ their original sizes. See Figure 1 for a comparison of dataset scales.

Table 3: Statistics of datasets included in MGB. We use suffix '-MGB' to denote the newly curated datasets with textual attributes. 'BoW' denotes features encoded using a bag-of-words model, 'one-hot' denotes features encoded as one-hot embedding of node counts, and 'BERT' denotes features extracted from raw text using Sentence-BERT (Reimers & Gurevych, 2019).

| Datasets | Nodes | Edges | Scale | Relationships | Train/val/test | Features | Classes | Has Text |
|---|---|---|---|---|---|---|---|---|
| ACM | 3,025 | 2,210,761 29,281 | Small | Paper-Subject-Paper(PSP) Paper-Author-Paper (PAP) | 600/300/2125 | 1,870 (BoW) | 3 | ✗ |
| IMDB | 4,780 | 98,110 21,018 | Small | Movie-Actor-Movie (MAM) Movie-Director-Movie (MDM) | 300/300/2687 | 1,232 (BoW) | 3 | ✗ |
| DBLP | 4,057 | 11,113 5,000,495 6,776,335 | Small | Author-Paper-Author (APA) Author-Paper-Conf-Paper-Author (APCPA) Author-Paper-Term-Paper-Author (APTPA) | 800/400/2857 | 334 (BoW) | 4 | ✗ |
| Amazon | 7,621 | 266,237 1,104,257 16,305 | Small | Also-view (IVI) Also-bought (IBI) Bought-together (IOI) | 80/200/7341 | 2,000 (BoW) | 4 | ✗ |
| Freebase | 3,492 | 254,702 8,404 10,706 | Small | Movie-Actor-Movie (MAM) Movie-Director-Movie (MDM) Movie-Writer-Movie (MWM) | 60/1000/1000 | 3492 (one-hot) | 3 | ✗ |
| Amazon-fraud | 11,994 | 175,608 3,566,479 1,036,737 | Medium | User-Product-User (UPU) User-Star_rate-User (USU) User-Text-User (UVU) | 2388/1194/8362 | 25 (handcraft) | 2 | ✗ |
| Yelp-fraud | 45,954 | 49,315 573,616 3,402,743 | Large | Review-User-Review (RUR) Review-Time-Review (RTR) Review-Star_rate-Review (RSR) | 9191/4595/32168 | 32 (handcraft) | 2 | ✗ |
| ACM-MGB | 10,041 | 14,916,901 151,014 100,820,841 | Medium | Paper-Subject-Paper(PSP) Paper-Author-Paper (PAP) Paper-Term-Paper (PTP) | 2007/1002/7032 | 768 (BERT) | 5 | ✓ |
| IMDB-MGB | 4,573 | 93,039 19,327 | Small | Movie-Actor-Movie (MAM) Movie-Director-Movie (MDM) | 912/454/3207 | 768 (BERT) | 5 (multi-label) | ✓ |
| DBLP-MGB | 11,081 | 266,557 17,589 110,079,981 | Medium | Paper-Author-Paper (PAP) Paper-Paper (PP) Paper-Author-Term-Author-Paper (PATAP) | 2215/1107/7759 | 768 (BERT) | 3 | ✓ |
| Amazon-MGB | 35,000 | 36,855 102,276 3,950 | Large | Also-view (IVI) Also-bought (IBI) Bought-together (IOI) | 7000/3500/24500 | 768 (BERT) | 7 | ✓ |

**Plug-and-Play Implementations** All updated datasets, along with existing commonly-used datasets, were organized into a standardized format using PyTorch (Paszke et al., 2019). Dataset statistics are provided in Table 3, and preprocessing methods, such as graph normalization, are offered in a plug-and-play fashion. We release all data, code, and documentations publicly available, allowing researchers to customize datasets according to their specific needs (see links in Appendix A) .

### 4.3 EVALUATION METRICS

**Node Classification & Clustering** For node-level tasks, we adopted node classification and clustering tasks, following previous works. Specifically for the IMDB-MGB dataset, we corrected the original single-label three-class classification task to a multi-label five-class classification task by adding categories 'Thriller' and 'Romance'. This enhancement increases the credibility and difficulty of the node classification task. For evaluation metrics, we used Macro-F1 and Micro-F1 for node classification, and Accuracy and Normalized Mutual Information (NMI) for node clustering.

**Edge Prediction & Classification** To address the previous neglect of edge-level tasks in existing works, we proposed two metrics to evaluate model performance on edge tasks. Firstly, we employed the edge prediction task, which involves determining whether a given edge exists in the graph, constructed with random negative sampling to generate negative samples. This approach helps the model learn to distinguish between true edges and randomly sampled negative edges. For each positive edge, we sample five corresponding negative edges. To assess performance on this task, we utilized the area under the ROC curve (AUC-ROC) and the Precision-Recall curve (AUC-PR).

Innovatively, we introduced a novel edge classification task based on multiple meta-paths within multiplex graphs. This task requires the model to accurately classify existing edges in the test set, associating them with specific meta-paths. Taking ACM-MGB as an example, given an existing edge, the model needs to determine whether it belongs to one of the meta-paths 'PSP, PAP, PTP'. We used F1 score as the classification metric to evaluate performance on this task, providing insights into the model's ability to classify edges based on their meta-path associations.

## 5  FAIR COMPARISON UNDER MGB

Based on the proposed MGB, we conducted a more comprehensive comparison of existing methods. The main results are presented in Table 4. Since our proposed IMDB-MGB dataset is multi-labeled, we did not report the clustering task designed for single-labeled datasets. Additional results on edge-level tasks and two binary classification anomaly detection datasets (Dou et al., 2020) can be found in Appendix E. These experiments yielded several new observations:

Table 4: Fair comparison results under our proposed Multiplex Graph Benchmark (MGB), where higher value is better. Colored are the top first, second, and third results. *E2E* denotes end-to-end methods, *SS* denotes self-supervised methods. OOM indicates out-of-memory error.

(a) Node classification results on proposed MGB datasets with textual attributes.

| Methods | | ACM-MGB | | IMDB-MGB | | DBLP-MGB | | Amazon-MGB | |
| | | Macro-F1 | Micro-F1 | Macro-F1 | Micro-F1 | Macro-F1 | Micro-F1 | Macro-F1 | Micro-F1 |
|---|---|---|---|---|---|---|---|---|---|
| *E2E* | GCN | 58.21±0.9 | 56.31±1.3 | 50.72±1.5 | 55.03±0.6 | 95.04±1.4 | 96.68±0.7 | 87.51±6.1 | 88.00±5.6 |
| | GAT | 28.30±7.5 | 34.54±8.1 | 38.51±9.8 | 39.70±10.3 | 57.09±6.6 | 64.24±9.7 | OOM | |
| | HAN | 37.08±10.1 | 41.56±8.0 | 34.64±8.0 | 36.26±8.4 | 68.53±10.4 | 76.44±10.0 | OOM | |
| *SS* | MNE | 47.52±0.8 | 48.01±0.6 | 38.64±0.6 | 42.34±0.6 | 43.76±0.4 | 55.89±0.9 | OOM | |
| | DMGI | 48.66±4.3 | 56.52±0.7 | 28.47±0.3 | 36.59±0.3 | 28.19±1.2 | 64.67±0.5 | 13.02±0.2 | 14.38±0.1 |
| | HDMI | 56.32±1.6 | 54.72±1.3 | 36.33±4.0 | 43.47±4.0 | 81.82±3.4 | 87.68±2.0 | 76.83±2.1 | 76.86±2.0 |
| | CKD | 60.05±0.3 | 58.09±0.6 | 39.61±0.2 | 47.41±0.6 | 88.41±0.8 | 91.42±0.6 | 86.89±0.9 | 86.88±0.9 |
| | MGDCR | 57.25±1.0 | 55.90±0.8 | 35.36±0.7 | 44.03±0.4 | 66.20±8.5 | 78.78±4.5 | 79.31±0.7 | 79.36±0.7 |
| | DMG | 48.55±4.6 | 51.32±3.6 | 17.91±0.9 | 40.03±0.6 | 36.00±1.8 | 66.68±0.6 | 28.75±5.1 | 31.02±4.0 |

(b) Node clustering results on proposed MGB datasets with textual attributes.

| Methods | | ACM-MGB | | DBLP-MGB | | Amazon-MGB | |
| | | Accuracy | NMI | Accuracy | NMI | Accuracy | NMI |
|---|---|---|---|---|---|---|---|
| *E2E* | GCN | 37.73±1.9 | 5.42±4.0 | 64.64±0.1 | 0.42±0.2 | 14.33±0.0 | 0.09±0.0 |
| | GAT | 44.41±4.0 | 20.66±7.3 | 63.63±1.6 | 2.60±1.7 | OOM | |
| | HAN | 41.61±5.2 | 15.40±9.1 | 64.89±0.6 | 1.74±2.2 | OOM | |
| *SS* | MNE | 35.47±0.0 | 0.12±0.0 | 64.52±0.0 | 0.10±0.0 | OOM | |
| | DMGI | 48.76±3.7 | 25.56±2.2 | 63.40±0.9 | 0.31±0.2 | 14.40±0.1 | 0.17±0.1 |
| | HDMI | 38.68±2.7 | 8.10±7.3 | 64.58±0.1 | 0.29±0.2 | 14.37±0.0 | 0.11±0.0 |
| | CKD | 46.52±2.5 | 27.81±6.2 | 64.50±0.0 | 0.07±0.0 | 14.30±0.0 | 0.05±0.0 |
| | MGDCR | 38.25±5.0 | 6.00±7.8 | 64.45±0.0 | 0.06±0.0 | 14.41±0.0 | 0.16±0.1 |
| | DMG | 37.33±1.8 | 4.63±4.8 | 64.57±0.0 | 0.25±0.1 | 14.37±0.0 | 0.15±0.0 |

(c) Edge prediction & classification results on proposed MGB datasets with textual attributes.

| Methods | ACM-MGB | | | IMDB-MGB | | | DBLP-MGB | | | Amazon-MGB | | |
| | AUC-ROC | AUC-PR | F1 | AUC-ROC | AUC-PR | F1 | AUC-ROC | AUC-PR | F1 | AUC-ROC | AUC-PR | F1 |
|---|---|---|---|---|---|---|---|---|---|---|---|---|
| GCN | 77.91±0.2 | 40.48±0.7 | 57.84±0.3 | 58.89±1.0 | 22.71±2.8 | 51.58±5.9 | 47.71±7.1 | 16.61±3.1 | 36.05±3.0 | 54.78±3.3 | 18.68±1.9 | 41.40±4.4 |
| GAT | 66.74±5.9 | 24.91±5.2 | 53.35±5.3 | 54.60±2.7 | 30.73±16.4 | 48.52±1.0 | 61.30±5.9 | 24.62±3.3 | 36.35±3.6 | OOM | | |
| HAN | 68.20±4.8 | 33.26±8.9 | 55.76±7.4 | 55.90±2.4 | 20.50±3.1 | 53.03±3.2 | 63.17±3.6 | 27.19±2.9 | 38.98±3.7 | OOM | | |
| MNE | 63.53±1.3 | 24.16±1.4 | 51.42±6.5 | 52.12±0.8 | 17.94±0.2 | 52.12±0.8 | 55.38±1.2 | 20.83±0.6 | 37.23±1.2 | OOM | | |
| DMGI | 77.60±1.6 | 48.10±4.3 | 61.77±0.8 | 54.55±0.9 | 19.96±1.3 | 53.71±0.3 | 54.52±0.2 | 19.20±0.2 | 39.77±0.9 | 52.36±0.4 | 19.01±0.4 | 34.49±0.6 |
| HDMI | 79.98±1.3 | 44.87±2.0 | 68.75±0.3 | 56.75±2.2 | 21.20±1.4 | 51.82±2.5 | 64.60±4.3 | 27.47±2.8 | 43.18±1.6 | 54.83±1.9 | 19.80±1.7 | 36.80±2.3 |
| CKD | 80.24±0.2 | 46.69±1.1 | 64.00±1.4 | 64.55±1.2 | 27.07±0.9 | 58.28±0.4 | 67.62±0.8 | 29.14±0.7 | 45.84±1.0 | 56.38±1.5 | 20.14±1.3 | 41.05±1.5 |
| MGDCR | 79.02±1.1 | 43.66±1.2 | 67.72±0.4 | 63.94±0.6 | 27.58±0.8 | 60.07±1.0 | 80.66±2.6 | 43.22±4.1 | 49.04±1.3 | 59.52±0.7 | 23.30±0.5 | 43.08±1.2 |
| DMG | 75.25±3.6 | 35.52±4.5 | 55.94±1.4 | 53.07±2.0 | 18.88±1.0 | 50.09±1.4 | 56.18±5.2 | 20.01±2.5 | 32.67±3.4 | 52.91±2.0 | 18.19±1.3 | 35.64±2.6 |

**Observation 4. Lower performance on new challenging MGB datasets.** As shown in Table 4, the performance of various methods on node classification, clustering, and edge tasks consistently decreases on the newly proposed MGB datasets, indicating significant room for improvement for existing methods. For instance, on the original ACM dataset (Wang et al., 2019b), baseline methods reportedly achieved node classification performance of over 80%. However, on our ACM-MGB dataset, which features a larger number of nodes, meta-paths, label space, and richer textual features, the best result was only 60.05/58.09 achieved by CKD (Wang et al., 2022a). Additionally, compared to

the issue of closely clustered performances of various methods on previous datasets, the performance gaps on the MGB datasets have largely increased. For instance, methods like DMGI (Park et al., 2020) and DMG (Mo et al., 2023b) even exhibit underfitting on the node classification tasks. The increased complexity of the proposed datasets makes them more challenging for the models, which offers ample opportunities for future research.

Table 5: Ablation study on the difficulty of MGB datasets, where results are reported in Macro-F1 score of node classification.

| Description | GCN | HDMI | MGDCR | std($\sigma$) |
|---|---|---|---|---|
| Amazon-MGB (full text embedding, 3 meta-paths, 7 classes) | 87.51 | 76.83 | 79.31 | 5.59 |
| partial text+BoW embedding, 3 meta-paths, 7 classes | 64.08 | 53.95 | 52.82 | 6.20 |
| full text embedding, 1 meta-path (only IBI), 7 classes | 87.97 | 80.64 | 79.75 | 4.51 |
| full text embedding, 3 meta-paths, 4 classes | 91.55 | 83.36 | 84.69 | 4.40 |

To further validate the improved difficulty of our proposed MGB datasets, we conducted an ablation study on the Amazon-MGB dataset. The results are shown in the Table 5. First, we reduced the quality of the feature embeddings by using only product titles as raw text and a simple bag-of-words model for encoding. When comparing this with the standard Amazon-MGB dataset, which uses product titles, brands, and descriptions as input and employs a BERT encoder, we found that the richer text-based embeddings do indeed improve model performance significantly. Next, we reduced the difficulty of the dataset by lowering the number of meta-paths and label classes. We observed that by simplifying the dataset, the performance of various methods improved, but the differentiation ($\sigma$) between methods also decreased. This demonstrates that overall, our new MGB datasets have led to a decrease in the performance of existing methods, but they also provide a better distinction between the actual performance gaps of the models.

**Observation 5. Rethinking possible designs for future multiplex graph models.** Table 4b illustrates that all methods produce nearly random outcomes for clustering results on the DBLP-MGB and Amazon-MGB datasets. This phenomenon may stem from the relatively shallow depth of existing methods and the increased sparsity and complexity of new datasets, which leads to an inability to learn effective clustering

Table 6: Results for node clustering and edge-level tasks of a Graph Transformer.

| Dataset | Acc | NMI | AUC-PR | F1 |
|---|---|---|---|---|
| IMDB-MGB | NA | NA | 33.25±0.5 | 60.09±3.1 |
| DBLP-MGB | 95.66±2.3 | 83.91±2.4 | 43.35±0.8 | 49.87±0.7 |
| Amazon-MGB | 89.64±1.7 | 82.40±0.9 | 42.01±0.7 | 49.64±0.4 |

features. To validate this hypothesis, we tested a simple end-to-end Graph Transformer model (see detailed implementation in Appendix D and complete results on Graph Transformer in Appendix E.4). As shown in Table 6, its node clustering performance significantly improved. This suggests that the random results obtained by other methods are due to inadequate learning rather than errors in the datasets or experimental settings. Further t-SNE (van der Maaten & Hinton, 2008) plots in Appendix E.3 also indicate the insufficiency of existing methods in learning distinguishable representations on the MGB datasets.

Similarly, Table 4c reveals unsatisfactory results for the newly proposed edge-level tasks, given that the models were not explicitly optimized for such objectives. Nonetheless, certain methods, such as MGDCR (Mo et al., 2023a) and CKD (Wang et al., 2022a), showcase relatively acceptable performances on the ACM-MGB and DBLP-MGB datasets. Furthermore, as illustrated in Table 6, more advanced models like the Graph Transformer with global attention exhibit considerable enhancements compared to existing methods. Thus, in conjunction with Observation 3, this observation underscores the potential for designing multiplex graph models better suited to large-scale datasets with intricate textual features, particularly models supporting global attention and larger-scale end-to-end architectures.

## 6 CONCLUSIONS

**Limitations and Future Directions** A primary limitation of MGB is its current scale. Constrained by the size of the original data and available computational resources, the largest dataset, Amazon-MGB, contains 35,000 nodes. In contrast, existing large-scale graph benchmark OGB (Hu et al., 2020) support millions of nodes, presenting more challenging scenarios. Additionally, balancing the number

and weights of edges for different relationships (Mo et al., 2023b) requires further investigation. In the future, we plan to expand MGB by incorporating more large-scale datasets and state-of-the-art models.

**Conclusion** In this paper, we identified and addressed significant issues in existing methods, datasets, and evaluation metrics in the field of multiplex graph learning. Initially, we conducted a fair comparison under the current settings. Preliminary experimental results highlighted inconsistencies in previous method comparisons, near-saturation of performance on existing low-differentiation datasets, and the potential advantages of end-to-end methods in the current setting. To further address the inadequacies of existing datasets and evaluation metrics, we proposed MGB, a comprehensive multiplex graph benchmark that currently includes 10 unified baseline implementations, 11 diverse datasets, and 4 comprehensive evaluation tasks. Further experiments on MGB revealed the limitations of existing models when faced with more challenging data and tasks, prompting us to rethink the design of future multiplex graph models. Specifically, models with global attention and stronger expressive power in end-to-end solutions show promise.

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

## A  DATASET DOCUMENTATION AND INTENDED USE

### A.1  DATASET LICENSES AND DOWNLOAD LINKS

We provide the dataset licenses and downloadable links below. In this paper, we use the most acclaimed versions of the ACM, IMDB, and DBLP datasets from the paper "Heterogeneous Graph Attention Network"(Wang et al., 2019b), the Amazon dataset from "Unsupervised Attributed Multiplex Network Embedding"(Park et al., 2020), and the Freebase dataset from "Self-Supervised Heterogeneous Graph Neural Network with Co-Contrastive Learning" (Wang et al., 2021). We also credit the original data collectors and annotators.

- **ACM:** Unknown License. The data were collected and processed by (Wang et al., 2019b), and can be downloaded from here.
- **IMDB:** CC0 v1.0 License. The raw data is an open-source dataset from Kaggle [3]. HAN (Wang et al., 2019b) processed it into a version containing 4,780 nodes, which can be found here.
- **DBLP:** Unknown License. The raw data were collected by (Gao et al., 2009) and is available at [4]. HAN (Wang et al., 2019b) provided a version containing 4,057 nodes, which can be found here.
- **Amazon:** MIT License. The authors of (Ni et al., 2019) retrieved reviews and metadata from Amazon[5]. The widely used version from (Park et al., 2020) includes a 4-category subset from the raw data, which can be downloaded here.
- **Freebase:** CC BY License. The raw data were collected by (Yang et al., 2020). We use the most commonly used version from (Wang et al., 2021), available here.
- **Amazon-Fraud, Yelp-Fraud:** Apache License 2.0. The collectors are (Dou et al., 2020), and the datasets can be found here.
- **ACM-MGB, IMDB-MGB, DBLP-MGB, Amazon-MGB:** MIT License. We provide the raw data and the processed data of our proposed MGB dataset from here.

As authors, we confirm the data licenses as indicated above and we bear all responsibility in case of violation of rights.

To ensure a unified and reproducible fair comparison, we have uploaded all datasets at `https://drive.google.com/file/d/1LsJPsfr5tB2zK8ELlATxomJn687ToohX/view?usp=drive_link`. Simply place the datasets in the ./data folder under the MGB code directory for automatic execution.

---

[3]https://www.kaggle.com/datasets/karrrimba/movie-metadatacsv

[4]http://web.cs.ucla.edu/~yzsun/data/

[5]https://cseweb.ucsd.edu/~jmcauley/datasets/amazon/links.html

## A.2 MAINTENANCE PLAN

To provide up-to-date, robust, and reliable multiplex graph datasets for academic purposes, we will update and supplement the datasets based on the latest advancements in the field and community feedback. We will continuously maintain the git repository at `https://anonymous.4open.science/r/multiplex-F150`, including more baseline methods and datasets. We also provide a website `https://mg-benchmark.github.io/Multiplex-graph-Benchmark/`, which includes documented tutorials and running examples. In the future, we plan to add a Leaderboard feature to facilitate open competition and comparison.

## B RELATED WORKS

In this section, we briefly introduce the multiplex graph baseline methods reproduced in this paper.

**Graph Neural Networks (GNNs)**, including GCN (Kipf & Welling, 2016) and GAT (Veličković et al., 2017), encode each relationship in multiplex graphs separately and use an average readout to obtain the final embeddings.

**Heterogeneous Graph Attention Network (HAN) (Wang et al., 2019b)** proposes hierarchical attention mechanisms at both the node and relationship levels, trained end-to-end. Node-level attention learns the importance between a node and its meta-path-based neighbors, while relationship-level attention learns the importance of different meta-paths.

**Multiplex Network Embedding (MNE) (Zhang et al., 2018)** assigns each node a common and private embedding, allowing multiple relationships to be learned jointly using the Skip-gram algorithm (Mikolov et al., 2013).

**Deep Multiplex Graph Infomax (DMGI) (Park et al., 2020)** learns embeddings for multiplex graphs by maximizing the mutual information between local graph patches and the global graph representation. DMGI introduces a consensus regularization framework to minimize disagreements among relation-type-specific node embeddings and employs a universal discriminator to differentiate true sample pairs, regardless of relation types.

**High-order Deep Multiplex Infomax (HDMI) (Jing et al., 2021)** is similar to DMGI but also considers joint supervision signals, incorporating both extrinsic and intrinsic mutual information through high-order mutual information.

**Heterogeneous Graph with Co-contrastive Learning (HeCo) (Wang et al., 2021)** employs a cross-view contrastive mechanism. Specifically, it proposes two views of a heterogeneous information network to learn node embeddings, capturing both local and high-order structures. Additionally, HeCo introduces cross-view contrastive learning and a view mask mechanism to extract positive and negative embeddings from the two views.

**Collaborative Knowledge Distillation (CKD) (Wang et al., 2022a)** models the knowledge in each meta-path with two granularities: regional and global knowledge. It learns meta-path-based embeddings by collaboratively distilling knowledge from intra-meta-path and inter-meta-path perspectives simultaneously.

**Multiplex Graph Representation Learning via Dual Correlation Reduction (MGDCR) (Mo et al., 2023a)** addresses the problem of noisy information by investigating intra- and inter-graph decorrelation losses. MGDCR also designs a simple pretext task to eliminate the need for negative sampling in contrastive learning.

**Disentangled Multiplex Graph Representation Learning (DMG) (Mo et al., 2023b)** disentangles common and private information in multiplex graphs and designs a contrastive constraint to preserve complementarity while removing noise from private information.

## C DETAILS OF MGB

We implement the proposed benchmark and models set using the PyTorch (Paszke et al., 2019) and PyTorch-Geometric (Fey & Lenssen, 2019) frameworks. The experiments are conducted on Nvidia GeForce 2080Ti (11GB VRAM) and 3090Ti (24GB VRAM) GPUs.

## C.1 HYPERPARAMETERS FOR BASELINES

In this section, we present detailed hyperparameters for each baseline method. Initially, we provide general hyperparameters applicable to all baselines across all datasets. Unless explicitly specified by the original authors, these hyperparameters will be fine-tuned using a grid search. Note that for each method, not all hyperparameters are utilized.

- Training hyperparameters
  - random seeds: [0, 1, 2, 3, 4]
  - train epochs: 200
  - train learning rate: [0.0001, 0.001, 0.01]
  - weight decay: 1e-4
  - early stop patience: 20
  - test epochs: 100
  - test learning rate: 0.1
  - batch size: [64, 128]
- GNN encoder
  - hidden dimension: [32, 64, 128, 256, 512]
  - layers: [1, 2, 3, 4]
  - dropout: 0.1
  - isBias: True
  - activation: 'relu'
- MLP encoder
  - hidden dimension: [64, 128, 256]
  - layers: [1, 2, 3]
  - dropout: 0.1
  - isBias: True

We then specify hyperparameters for each baseline method. We attempt to use the original settings defined by the authors; however, for some experiments, reproducing results using the original parameters is not feasible. Therefore, the provided parameter options for grid search might be different from their original papers.

- GCN & GAT
  - refer to parameters of GNN encoder above
- HAN
  - refer to parameters of GNN encoder above
  - nheads: [1,2,4]
- MNE
  - p: [1, 2]
  - q: [0.5, 1]
  - walk_length: [10, 20]
  - context_size: [5, 10]
  - walks_per_node: [10, 100]
  - num_negative_samples: 1
- DMGI
  - reg_coef: [0.001, 0.01, 0.1]
  - sup_coef: [0.1, 0.2]
  - margin: [0.1, 0.3]
  - nheads: [1, 2, 4]
- HDMI

- coef_layers: [[1, 2, 0.001]]
- coef_fusion: [[0.01, 0.1, 0.001]]

- HeCo
  - tau: [0.5, 0.7, 0.9]
  - lam: 0.5
  - feat_drop: [0.1, 0.3, 0.5]
  - attn_drop: [0.1, 0.3, 0.5]

- CKD
  - negative_cnt: 5
  - topk: [10, 20, 30]
  - sample_times: 1
  - neigh_por: 0.6
  - global_weight: [0.05, 0.1, 0.15]

- MGDCR
  - lambda_intra: 0.01
  - lambda_inter: 0.0001
  - w_intra: [0.1, 1]
  - w_inter: 1

- DMG
  - c_dim: 8
  - p_dim: 2
  - phi_hidden_size: 256
  - phi_num_layers: 2
  - alpha: [0.02, 0.06, 0.1]
  - beta: [0.05, 0.8, 1]
  - lambda: [0.05, 0.5, 3]
  - tau: [0.5, 0.7]
  - neighbor_num: 300
  - sample_neighbor: [30, 50]
  - sample_num: 50
  - inner_epochs: 10

## D  A SIMPLE GRAPH-TRANSFORMER IMPLEMENTATION

In Section 5, we mentioned that we implemented a simple end-to-end graph transformer model, which achieved significant performance improvements on the MGB datasets. In this section, we will introduce the implementation details of this model. The code will also be open-sourced along with the implementations of other baseline methods.

### D.1  MULTIPLEX GRAPH POSITIONAL ENCODING

We interpret multiplex graph positional encoding from two perspectives. One perspective involves the node's position within the entire graph, known as the absolute position. This assigns a unique identification to each node, representing its global location in the overall graph. Another perspective involves the node's relative position concerning its neighbors and substructure, referred to as relative position. This type of position information is valuable for capturing nodes' local relationships.

**Laplacian-PE as absolute positional encoding**    From the global perspective, we introduce eigenvectors of the graph Laplacian as the absolute positional encoding. Specifically, the eigenvectors are computed by the factorization of the graph Laplacian matrix:

$$\mathbf{L}_r = \mathbf{I} - \mathbf{D}^{-\frac{1}{2}} \mathbf{A}_r \mathbf{D}^{-\frac{1}{2}} = \mathbf{U}_r^T \Lambda_r \mathbf{U}_r, \tag{1}$$

where $\mathbf{L}_r$ and $\mathbf{A}_r$ represent the graph Laplacian and adjacency matrix of multiplex graph $\mathcal{G}_r$, respectively. $\mathbf{D}$ is the graph degree matrix, $\mathbf{I}$ is the identity matrix, and $\Lambda_r$ and $\mathbf{U}_r$ are the eigenvalues and eigenvectors of the $r$-th meta-path, respectively.

With the eigenvectors $\mathbf{U}_r$, we select $k$-smallest values and the Laplacian PE of node $v$ is defined as:

$$\mathbf{u}_{v|r} = [\mathbf{U}_{v1,r}, \mathbf{U}_{v2,r}, ..., \mathbf{U}_{vk,r}] \in \mathbb{R}^k. \tag{2}$$

The input node embeddings are the concatenation of the feature matrix and the Laplacian PE:

$$\mathbf{H}_r = \mathbf{X}_r \parallel \mathbf{U}_r[:,:k]. \tag{3}$$

For multiplex graphs with multiple meta-paths (i.e., multiple adjacency matrices), we pre-compute each meta-path's Laplacian PE and combine it with the feature matrix. Consequently, each meta-path's feature matrix contains both meta-path-specific attributive and structural information.

**RandomWalk-PE as relative positional encoding**    From the local perspective, nodes' relative positions or distances from each other also play a vital role. To model such relationships, we leverage random walk as the relative positional encoding of node pairs, acting as a soft inductive bias. The $p$-steps random walk matrix $\Phi_r$ of $r$-th meta-path is defined:

$$\Phi_r = (\mathbf{I} - \beta \mathbf{L}_r)^p, \tag{4}$$

where $\beta$ controls the amount of diffusion value between $[0.25, 0.50]$, and $p$ is the number of steps in the random walk. The entry $\Phi_r[i, j]$ indicates the possibility of node $i$ reaching node $j$ after a $p$-step random walk, representing the proximity relation in the graph.

### D.2    GRAPH SERIALIZATION

Inputting graph data into a transformer encoder poses challenges due to the computational constraints of self-attention. Transformers have a sequence length limit, whereas graph data often comprises thousands of nodes, making serialization of the entire graph impractical.

To address this issue, we propose an efficient and dynamic graph serialization strategy utilizing the properties of multiplex graphs. Specifically, for a node $v$, we sample its first-order neighbors $\mathcal{N}_{v|r}$ under each $\mathcal{G}_r$. If $|\mathcal{N}_{v|r}|$ exceeds the predefined maximum sequence length $L$, we randomly sample from it. This way, for node $v$ under different multiplex graph meta-paths, its corresponding ego-graph sequences $\mathcal{S}_{v|r} = \{v, u_1, \ldots, u_i, \ldots | u_i \in \mathcal{N}_{v|r}\}$ are generated, with $v$ always as the first node in the sequence. This approach allows us to dynamically and efficiently sample and serialize large graphs.

### D.3    TRANSFORMER ENCODER

For $r$-th meta-path of the multiplex graph, we inject the relative PE $\Phi_r$ into the multi-head self-attention mechanism to bias the attention score with node relative relations:

$$\text{Self-attn}(\mathbf{H}_r) = \text{softmax}\left(\frac{\mathbf{Q}_r \mathbf{K}_r^T}{\sqrt{d}} + \Phi_r\right)\mathbf{V}_r, \tag{5}$$

$$\mathbf{Q}_r = \mathbf{H}_r \mathbf{W_Q}, \ \mathbf{K}_r = \mathbf{H}_r \mathbf{W_K}, \ \mathbf{V}_r = \mathbf{H}_r \mathbf{W_V}, \tag{6}$$

where $\mathbf{W}$ is learnable linear projection, and $d$ is the dimension of $\mathbf{Q}_r$. Normally there will be multi-head attention so that each head can comprehend different aspects of information. We will omit that for simplicity of presentation. In this way, the self-attention of graph transformer encoder considers both the distance of node's feature space and relative positions.

After that, the hidden embeddings are passed into a series of skip-connection, normalization, and feed-forward networks (FFN), which combined together as a transformer encoder:

$$\hat{\mathbf{H}}_r = \text{Norm}(\mathbf{H}_r + \text{Self-attn}(\mathbf{H}_r)), \tag{7}$$

$$\mathbf{Z}_r = \text{Norm}(\hat{\mathbf{H}}_r + \text{FFN}(\hat{\mathbf{H}}_r)). \tag{8}$$

Multiple layers of such transformer encoder can be stacked to get deeper representation, which is also omitted for simplicity. For multiplex graphs, we initialize different parameter sets for each meta-path. And we average the output of the last encoder layer $\mathbf{Z}_r$ of each meta-path $\mathcal{G}_r$ as the final embedding:

$$\mathbf{Z} = \frac{1}{R} \sum_{r=1}^{R} \mathbf{Z}_r. \tag{9}$$

Then the aggregated node embeddings are passed into a task-related network for downstream tasks.

## E ADDITIONAL EXPERIMENTS

### E.1 RESULTS ON AMAZON-FRAUD AND YELP-FRAUD DATASETS

Table 7: The results of compared baseline methods on two fraud datasets.

(a) Results on Amazon-Fraud.

| Methods | Amazon-Fraud | | | | | | |
| | Node Classification | | Node Clustering | | Edge | | |
| | Macro-F1 | Micro-F1 | Accuracy | NMI | AUC-ROC | AUC-PR | F1 |
|---|---|---|---|---|---|---|---|
| GCN | 77.56±0.4 | 95.65±0.0 | 80.48±0.5 | 2.29±0.0 | 84.18±0.3 | 59.95±0.6 | 52.02±0.1 |
| GAT | 48.02±0.4 | 91.36±3.5 | 83.04±6.8 | 15.23±6.5 | 69.08±5.5 | 29.26±7.2 | 45.38±2.8 |
| HAN | 57.04±11.1 | 93.76±1.0 | 88.25±8.8 | 20.71±12.3 | 68.84±5.3 | 27.61±4.8 | 48.34±4.3 |
| MNE | 90.87±0.1 | 97.90±0.0 | 63.27±0.0 | 1.47±0.0 | 77.06±0.4 | 48.26±1.6 | 50.26±0.3 |
| DMGI | 81.44±0.6 | 96.01±0.2 | 72.72±1.4 | 3.66±2.5 | 84.44±1.0 | 63.54±1.2 | 63.76±1.4 |
| HDMI | 80.46±1.0 | 95.93±0.2 | 69.73±0.8 | 2.91±0.1 | 84.73±0.3 | 64.84±0.4 | 65.78±0.4 |
| CKD | 87.12±0.1 | 97.02±0.0 | 58.44±3.1 | 4.36±2.5 | 79.79±0.5 | 53.77±0.6 | 56.03±0.3 |
| MGDCR | 76.11±0.9 | 95.28±0.1 | 91.78±1.3 | 23.58±1.7 | 83.44±0.1 | 61.00±0.2 | 60.46±0.7 |
| DMG | 76.53±2.0 | 95.41±0.3 | 75.16±7.5 | 4.32±2.5 | 85.71±0.1 | 65.47±0.2 | 63.25±0.9 |

(b) Results on Yelp-Fraud.

| Methods | Yelp-Fraud | | | | | | |
| | Node Classification | | Node Clustering | | Edge | | |
| | Macro-F1 | Micro-F1 | Accuracy | NMI | AUC-ROC | AUC-PR | F1 |
|---|---|---|---|---|---|---|---|
| GCN | 46.08±0.0 | 85.47±0.0 | 60.95±12.4 | 0.03±0.0 | 50.22±2.0 | 16.97±0.7 | 33.31±2.1 |
| GAT | | | out-of-memory | | | | |
| HAN | | | out-of-memory | | | | |
| MNE | 47.62±1.2 | 85.54±0.1 | 55.85±2.0 | 0.02±0.0 | 56.03±0.4 | 19.32±0.3 | 39.28±0.3 |
| DMGI | 53.56±1.1 | 85.86±0.1 | 52.87±0.9 | 0.02±0.0 | 60.87±0.7 | 22.36±0.7 | 41.24±0.8 |
| HDMI | 46.08±0.0 | 85.47±0.0 | 58.02±5.1 | 0.03±0.0 | 61.86±0.7 | 23.35±0.6 | 43.08±0.8 |
| CKD | 46.09±0.0 | 85.46±0.0 | 66.85±8.3 | 0.02±0.0 | 59.78±0.8 | 22.29±0.7 | 40.09±1.7 |
| MGDCR | 46.08±0.0 | 85.47±0.0 | 52.01±2.2 | 0.04±0.0 | 57.96±0.5 | 21.06±1.0 | 40.30±0.5 |
| DMG | 46.95±0.4 | 85.45±0.1 | 55.35±1.6 | 0.35±0.2 | 55.42±1.1 | 19.38±0.9 | 39.33±1.1 |

The results of all methods on the two anomaly detection datasets (Dou et al., 2020) are listed in Table 7.

### E.2 EDGE-LEVEL TASKS ON EXISTING DATASETS

In Table 8, we present the edge prediction and classification results on previously existing datasets. It can be observed that for edge-level tasks, self-supervised methods generally outperform end-to-end methods, contrasting with Observation 3 in Section 3. The primary reason is that the end-to-end methods we tested were trained on node tasks and were not specifically retrained for edge tasks. Consequently, these methods did not adapt well to the new tasks. In contrast, self-supervised methods learn more general representations, which better generalize across various downstream tasks. Therefore, the choice of model should consider the specific downstream tasks and the associated training costs. Self-supervised methods may require more resources but offer better generalization, while end-to-end methods might be more efficient for specific tasks if appropriately retrained.

Table 8: Edge prediction & classification results on existing datasets.

(a) Results on ACM, IMDB, and DBLP datasets.

| Methods | ACM | | | IMDB | | | DBLP | | |
|---|---|---|---|---|---|---|---|---|---|
| | AUC-ROC | AUC-PR | F1 | AUC-ROC | AUC-PR | F1 | AUC-ROC | AUC-PR | F1 |
| GCN | 56.84±4.4 | 24.21±3.3 | 58.09±0.2 | 54.14±0.1 | 19.17±0.2 | 51.74±0.4 | 59.16±0.7 | 22.30±0.2 | 40.18±0.2 |
| GAT | 57.03±1.2 | 20.40±1.8 | 58.48±0.2 | 57.98±2.1 | 24.69±1.4 | 52.28±0.7 | 55.06±0.9 | 20.44±0.7 | 38.05±1.2 |
| HAN | 57.66±1.4 | 22.15±2.0 | 57.59±0.2 | 54.29±1.2 | 21.56±1.3 | 52.95±1.4 | 54.35±0.4 | 18.76±0.5 | 37.24±1.2 |
| MNE | 67.76±0.4 | 32.44±0.3 | 67.97±0.3 | 62.48±0.1 | 25.92±0.1 | 57.50±0.2 | 56.95±0.1 | 18.89±0.1 | 39.72±0.1 |
| DMGI | 79.27±2.9 | 61.13±7.9 | 73.44±0.9 | 62.68±2.7 | 27.22±3.0 | 58.04±1.9 | 60.61±1.0 | 22.09±0.6 | 43.20±0.6 |
| HDMI | 60.53±1.5 | 29.46±1.5 | 68.02±0.8 | 73.07±0.2 | 40.17±0.3 | 67.38±0.9 | 65.94±0.3 | 25.34±0.2 | 43.71±0.4 |
| HeCo | NA | NA | NA | NA | NA | NA | 68.82±0.4 | 27.06±0.2 | 45.88±0.4 |
| CKD | 66.22±3.8 | 36.83±10.1 | 65.81±1.5 | 68.53±2.7 | 32.46±3.6 | 59.11±1.7 | 62.23±1.5 | 22.74±1.3 | 39.37±0.3 |
| MGDCR | 55.09±2.5 | 22.03±2.1 | 64.52±0.5 | 69.10±0.5 | 36.24±2.0 | 61.45±1.2 | 69.15±0.2 | 27.27±0.3 | 45.84±0.3 |
| DMG | 67.63±5.9 | 31.21±8.2 | 68.80±1.1 | 65.97±1.3 | 25.40±3.2 | 56.72±1.5 | 70.48±0.0 | 28.04±0.1 | 44.63±0.2 |

(b) Results on Amazon and Freebase datasets.

| Methods | Amazon | | | Freebase | | |
|---|---|---|---|---|---|---|
| | AUC-ROC | AUC-PR | F1 | AUC-ROC | AUC-PR | F1 |
| GCN | 59.86±0.6 | 19.12±0.3 | 37.90±1.0 | 70.24±0.2 | 52.68±0.1 | 40.32±0.0 |
| GAT | 72.86±3.7 | 32.84±5.9 | 44.74±3.6 | 70.10±0.1 | 51.49±0.1 | 40.33±0.1 |
| HAN | 55.93±1.4 | 20.25±0.9 | 35.19±0.5 | 69.30±0.4 | 39.45±0.4 | 40.29±0.1 |
| MNE | 69.44±0.3 | 32.88±0.3 | 49.23±0.1 | 74.77±0.0 | 41.64±0.0 | 42.03±0.1 |
| DMGI | 63.45±1.0 | 28.01±1.5 | 43.70±0.8 | 67.61±4.1 | 34.75±7.1 | 43.03±2.1 |
| HDMI | 82.85±0.8 | 53.00±2.1 | 56.94±1.0 | 73.32±0.3 | 52.29±0.4 | 51.56±0.6 |
| HeCo | NA | NA | NA | 74.36±1.4 | 42.51±2.6 | 49.36±0.9 |
| CKD | 79.06±2.3 | 39.59±4.0 | 44.39±1.8 | 69.01±1.9 | 40.29±3.1 | 43.73±1.8 |
| MGDCR | 70.00±0.0 | 39.12±0.0 | 53.21±0.1 | 70.82±2.7 | 33.57±7.4 | 41.11±0.7 |
| DMG | 76.16±2.6 | 30.87±3.4 | 49.95±1.0 | 71.95±2.6 | 49.46±3.8 | 47.53±0.8 |

Additionally, differences between methods are more pronounced with the AUC-PR metric compared to AUC-ROC. For example, MGDCR (Mo et al., 2023a) and DMG (Mo et al., 2023b) have similar AUC-ROC scores on the Freebase dataset, but their AUC-PR scores differ by about 16%. This is because AUC-PR is generally a more suitable performance metric for imbalanced datasets, as it focuses on the model's ability to predict the minority class (positive examples). AUC-ROC, on the other hand, may mask the model's deficiencies in predicting positive examples due to the large number of negative examples in the dataset. In our setup, with a 5:1 ratio of negative to positive edges, AUC-PR more accurately reflects model differences.

### E.3 NODE CLUSTERING VISUALIZATION

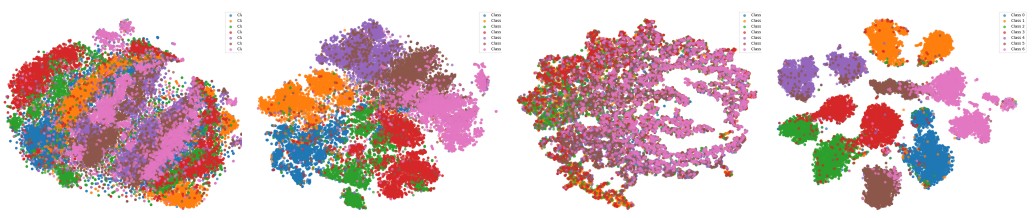

Figure 2: HDMI          Figure 3: CKD          Figure 4: DMG          Figure 5: Transformer

To provide a clearer understanding of the phenomenon observed in Table 4b, where existing methods exhibit near-random clustering performance on MGB datasets, we visualized the t-SNE plots for the HDMI (Jing et al., 2021), CKD (Wang et al., 2022a), DMG (Mo et al., 2023b), and Graph Transformer methods on the Amazon-MGB dataset. The plots in Figure 2-5 reveal the distinctiveness of the features learned by different models: The Graph Transformer model achieves large inter-class distances and small intra-class distances, indicating well-separated and cohesive clusters. CKD performs slightly worse, with smaller inter-class distances. HDMI shows confusion between different classes. DMG suffers from underfitting, failing to learn distinguishable features. The clustering performance observed in these visualizations also correlates with the node classification performance reported in Table 4a, suggesting that better clustering results tend to correspond with higher node classification metric.

### E.4 ADDITIONAL RESULTS ON GRAPH TRANSFORMER

In this subsection, we provide complete results of the Graph Transformer model over all tasks on all datasets mentioned in our paper. *It is worth noting that the Graph Transformer was introduced in Observation 5 to validate our hypothesis that model depth and capacity can enhance performance. It was not formally proposed by other researchers previously, and our implementation is a very preliminary product without special design for multiplex relationships. Therefore, it was not included as a baseline in our main body comparisons.* The main results and ablation study are listed in Table 9 and Table 10.

Table 9: Main results for Graph Transformer across different datasets.

| Dataset | Macro-F1 | Micro-F1 | Accuracy | NMI | AUC-ROC | AUC-PR | F1 |
|---|---|---|---|---|---|---|---|
| ACM | 87.18 | 87.19 | 87.63 | 63.83 | 64.17 | 40.98 | 63.07 |
| IMDB | 52.00 | 53.87 | 51.36 | 8.59 | 68.67 | 37.43 | 57.68 |
| DBLP | 89.95 | 90.97 | 88.27 | 69.27 | 72.51 | 34.96 | 47.82 |
| Amazon | 64.78 | 65.24 | 52.55 | 15.25 | 80.64 | 39.75 | 49.80 |
| Freebase | 50.55 | 52.60 | 51.88 | 10.81 | 79.18 | 36.89 | 46.03 |
| ACM-MGB | 63.12±1.3 | 61.15±1.6 | 58.46±2.6 | 43.50±1.0 | 82.58±0.6 | 54.01±0.6 | 69.49±1.2 |
| IMDB-MGB | 37.10±1.2 | 58.95±1.1 | NA | NA | 61.74±0.4 | 33.25±0.5 | 60.09±3.1 |
| DBLP-MGB | 96.02±0.1 | 97.24±0.1 | 95.66±2.3 | 83.91±2.4 | 72.96±0.7 | 43.35±0.8 | 49.87±0.7 |
| Amazon-MGB | 92.28±0.3 | 93.33±0.3 | 89.64±1.7 | 82.40±0.9 | 65.82±0.3 | 42.01±0.7 | 49.64±0.4 |

Table 10: Ablation study for Graph Transformer on ACM-MGB.

| | Macro-F1 | Micro-F1 | Accuracy | NMI |
|---|---|---|---|---|
| Graph Transformer | 63.12±1.3 | 61.15±1.6 | 58.46±2.6 | 43.50±1.0 |
| w/o absolute positional encoding | 61.80±1.9 | 60.10±2.1 | 55.75±1.6 | 42.27±1.9 |
| w/o relative positional encoding | 58.97±2.1 | 57.79±1.8 | 57.09±2.1 | 42.40±2.2 |
| single-head QKV, single encoder layer | 58.48±1.3 | 56.56±1.5 | 55.05±2.2 | 39.25±0.7 |

### E.5 RUNNING TIME ANALYSIS

We conducted the experiments on running efficiency (time required to train one epoch, in milliseconds) for a thorough analysis. The results are shown in Table 11. Notably, the CKD method, which involves sub-graph sampling for each meta-path, is the most time-consuming, even slower than the Graph Transformer. We will consider code-level optimizations to improve CKD's speed in the future. It's important to note that our tests were conducted on an NVIDIA GeForce RTX 3090 paired with an Intel(R) Xeon(R) CPU E5-2678 v3 @ 2.50GHz. The hardware setup may influence the results, and occasional load from other tasks running on the machine can also affect training speed. Therefore, these results should be only considered as preliminary comparisons.

Table 11: Training time per epoch in milliseconds.

| | ACM-MGB | IMDB-MGB | DBLP-MGB | Amazon-MGB |
|---|---|---|---|---|
| GCN | 2.33-14.21 | 4.34-8.42 | 9.63-24.46 | 87.10-287.46 |
| GAT | 35.32-40.69 | 28.34-33.16 | 133.23-261.47 | OOM |
| HAN | 123.47-138.07 | 23.33-30.49 | 167.93-556.01 | OOM |
| MNE | 475.12-869.91 | 185.41-571.92 | 327.64-1478.81 | OOM |
| DMGI | 3.80-19.15 | 5.98-46.62 | 17.85-64.23 | 102.65-358.44 |
| HDMI | 9.64-66.51 | 7.38-44.50 | 38.70-59.28 | 171.73-349.32 |
| CKD | 1879.45-6710.01 | 1010.37-3720.99 | 2105.74-3577.79 | 18581.42-43175.20 |
| MGDCR | 1.34-18.19 | 4.86-15.88 | 17.72-21.90 | 64.25-140.05 |
| DMG | 416.24-810.14 | 162.63-476.75 | 352.91-564.65 | 2501.44-5977.83 |
| Graph Transformer | 374.64-1619.01 | 223.56-276.41 | 856.91-2205.48 | 7109.85-22591.39 |

## F SOCIAL IMPACT

**Positive Impact** By providing a unified platform for comparing different methods, our work promotes rigorous and reproducible research, enabling fair and meaningful comparisons. This fosters

innovation and collaboration among researchers, leading to the development of more robust and expressive models. Enhanced models can be applied to various domains, such as social networks, bioinformatics, and recommendation systems, ultimately benefiting society by improving applications like fraud detection, personalized recommendations, and drug discovery.

**Negative Impact**  The focus on standardized benchmarks could potentially narrow the scope of research, as researchers might prioritize optimizing their models for these specific datasets rather than exploring broader or more diverse applications. Additionally, the increased computational resources required for large-scale benchmarks could exacerbate environmental concerns associated with high-energy consumption.

