# OpenReview forum: "Benchmarking and Rethinking Multiplex Graphs"
_ICLR.cc/2025/Conference — Submitted to ICLR 2025_

### Official Review · Reviewer_27VF · 2024-10-26

**Soundness:** 2
**Presentation:** 3
**Contribution:** 2
**Rating:** 3
**Confidence:** 4

**Summary:**

This paper presents a unified benchmark for multiplex graphs, which provides the implementations of several baseline methods, multiple multiplex graph datasets, as well as four node-level and edge-level evaluation tasks. The unified benchmark environment of the proposed MGB benchmark facilitates consistent and reproducible evaluation across different methods. Importantly, this paper aims to address the limitations of existing multiplex graphs by developing new multiplex graphs that are larger and more challenging than existing graphs, and include richer textual features than simplistic features available in existing datasets, such as one-hot encodings or bag-of-words features. Experimental results on both existing and newly proposed multiplex graphs show that for several baseline methods, the proposed multiplex graphs are more challenging than existing graphs, with larger performance variations among them.

**Strengths:**

S1. This paper presents a unified benchmark for multiplex graphs, which include the implementations of several baselines, multiple multiplex graphs, and four node-level and edge-level evaluation tasks. The unified environment facilitates more consistent and reproducible evaluation across different methods.

S2. This work develops four new multiplex graphs, which are more challenging, and have richer textual features than existing graphs. Experimental results on existing and newly proposed multiplex graphs show that the proposed graphs are more challenging for baseline methods, with larger performance variations among them, which can provide opportunities for further analysis of existing methods and development of more performant approaches.

S3. The writing is overall clear and the paper is easy to follow.

**Weaknesses:**

W1. The contributions of this work are rather limited and incremental.
* W1-a. This paper expands existing multiplex graph datasets like DBLP and IMDB, e.g., by adding more nodes and richer textual node features using BERT. While the expanded graphs are larger than the corresponding existing graphs, they are still relatively small (e.g., IMDB-MGB with ~4600 nodes and Amazon-MGB with ~35,000 nodes), and the improvements over existing datasets are not that significant. Importantly, there are several existing heterogeneous (multiplex) graphs that are larger than the graphs proposed in this paper. For example, OGB_MAG is a heterogeneous Microsoft academic graph with more than 1 million nodes; Freebase in HGBDataset has ~180k nodes, etc. Links to these graphs can be found [here](https://pytorch-geometric.readthedocs.io/en/stable/cheatsheet/data_cheatsheet.html#heterogeneous-datasets).

* W1-b. While it is claimed that the introduction of edge-level tasks is novel, many previous studies involving heterogeneous graphs like knowledge graphs perform edge-level evaluation, such as link prediction.

W2. Some observations made in the paper are inconsistent with results, or are not surprising.
* One observation says that current methods have nearly reached the performance ceiling on existing datasets. However, on existing datasets such as IMDB, Amazon, and Freebase, the classification performance of baselines methods are in the range of 40-60; thus it seems like there’s still large room for improvements for these existing datasets.
* Another observation is about end-to-end methods performing better than self-supervised methods. Given that end-to-end methods are directly optimized for the given task, it is expected for them to perform better than self-supervised methods, which are not directly optimized with respect to the given downstream task.

W3. This paper makes a distinction between multiplex graphs and heterogeneous graphs, but it is not clear what the differences are between them. In my understanding, they seem mostly the same, and methods developed for heterogeneous graphs could be applied to multiplex graphs. Thus, we have many heterogeneous graph datasets that are more complex and larger than those developed in this work, which limits the contributions of this work.

**Questions:**

Q1. How are the end-to-end methods applied to the clustering task? Since clustering is an unsupervised task, I’m not sure how methods can be applied in an end-to-end manner. What learning objective was used for their training?

Q2. For node clustering tasks, end-to-end methods like GCN cannot be optimized to predict the class label as in node classification. As such, their performance in node clustering is expected to be worse than that for node classification on the same dataset. However, there are several cases where the clustering performance of end-to-end methods is higher than the corresponding classification performance. This will need an explanation.

Q3. It is noteworthy that the best classification results on the newly proposed graphs are mainly achieved by the classical GCN method. Indeed, the gap between GCN and the second best method is quite significant (e.g., 50.72 vs 39.61, 95.04 vs 88.41). Given the increased complexity and size of the new datasets, this is an unexpected result. Could you provide some explanations for this?

---

### Official Review · Reviewer_pM7H · 2024-11-02

**Soundness:** 3
**Presentation:** 3
**Contribution:** 2
**Rating:** 6
**Confidence:** 3

**Summary:**

This paper proposes a new benchmark for multiplex graphs.
The authors first point out the issues of existing datasets, and then fix these issues by constructing new datasets with the same raw data.
Four new datasets with larger scale are constructed, and two new edge-level tasks are included.
The baselines include regular graph neural networks and self-supervised methods.

**Strengths:**

1. The paper is well written.
2. The experiments are comprehensive.

**Weaknesses:**

1. The motivation needs more justification.
2. More types of baselines are needed.

**Questions:**

1. Figure 1 is difficult to read. I recommend to separate it into two figures, namely number of nodes and edges, with x-axis and y-axis being previous and MGB, respectively. Therefore, an x=y line will clearly illustrate the increase of the scale.
2. In Section 4.2 paragraph “Expanding Dataset Scales”, how many nodes are there in the raw data? If not using the full graph provided by the raw data, how is the sampling of nodes (or edges) done?
3. In Section 4.3 paragraph “Edge Prediction & Classification”, for edge prediction, why not using Hits@K as it is used in [A]? Hits@K can be more useful and also more challenging in practical scenarios.
4. In Table 5, to better understand the results, is it possible to have a baseline (e.g. MLP) that only uses the text embedding? If so, it would be even better to include it as a common baseline for all tasks.
5. In Observation 5, why not include the graph transformer as a common baseline for all tasks?
6. As a benchmark paper, I would like to see more justifications why this benchmark is needed. For example, I am not fully convinced by the paragraph provided in line 108 and wondering why not just use relational GNNs. In this case, to show the necessity of GNNs tailored for multiplex graphs, it would be much better to include relational GNNs as a category of baseline for all experiments. Methods like RGCN [B] should work for heterogeneous graphs with a single entity type.

[A] https://ogb.stanford.edu/docs/linkprop/#ogbl-vessel

[B] Schlichtkrull, M., Kipf, T. N., Bloem, P., Van Den Berg, R., Titov, I., & Welling, M. (2018). Modeling relational data with graph convolutional networks. In The semantic web: 15th international conference, ESWC 2018, Heraklion, Crete, Greece, June 3–7, 2018, proceedings 15 (pp. 593-607). Springer International Publishing.

---

### Official Review · Reviewer_K2hv · 2024-11-03

**Soundness:** 1
**Presentation:** 3
**Contribution:** 1
**Rating:** 1
**Confidence:** 5

**Summary:**

The paper claims to be the first to benchmark multiplex graphs but it misses important existing prior art. To sum up, this work enriches four existing multiplex datasets and applies existing algorithms to them for standard edge classification and prediction, presenting it as a solid novelty.

**Strengths:**

1.) The paper aims to create a Multiplex Graph Benchmark, which is essentially running 10 existing techniques in a common hyperparameter setting, enriching four existing graph datasets with textual features, and using these for edge classification and prediction as well.

2.) Experiments show that the results on enriched datasets are poor in comparison to their original counterparts, depicting the level of difficulty that they offer.

**Weaknesses:**

1.) Important and very pertinent related work is missing [1]. The referred work established a benchmark on representation learning in multiplex graphs and also offers an information-fused taxonomy. So, to the best of my knowledge, this paper is not the first to do benchmarking in multiplex graph networks.

2.) GCN, GAT, and HAN are very old baselines that the authors have used solely for end-end learning. Works like [2], [3], and other recent methods need to be benchmarked and cannot be left for future work or to be mentioned in the limitation section.

3.) There are discrepancies in the values corresponding to train/val/test split vs the number of nodes for two datasets, namely, IMDB and Freebase. Why so?

4.) In Fig.1, why are there only 5 previous datasets shown? Either Freebase dataset should be omitted, or the Amazon-fraud and Yelp-fraud datasets should be added. Practically, the figure can be omitted as it just reports the number of nodes and edges in a dataset, which can be easily conveyed via a Table and has already been done in Table 3.

5.) Multiplexity and heterogeneity share a significant overlap, and I firmly believe that this intersection should have been duly incorporated in this paper. Please refer to works like the heterogeneous graph benchmark (HGB).

6.) Another recent work [4] should be presented in the section on self-supervised methods.

7.) Constraining all existing algorithms to run for the same number of epochs, having weight decay and learning rate, is not a correct step. Algorithms are different, and so do their hyperparameters. The correct way would have been to tune existing algorithms extensively and report the best hyperparameters and the corresponding results.

8.) I am not convinced by the explanation given for the poor performance on enriched datasets w.r.t. existing datasets. If the label space and meta paths are increased, so are the number of nodes and the textual content, which should have favoured better performance. It needs to be thoroughly investigated.

9.) Applying existing techniques and enriched data to standard edge classification and prediction tasks is not a novelty. The novelty would have been understood if existing methods had been custom-tailored for this new purpose.

*************************************
*************************************
References:

[1] Representation Learning in Multiplex Graphs: Where and How to Fuse Information? Computational Science – ICCS 2024. ICCS 2024. Lecture Notes in Computer Science, vol 14837. Springer, Cham. https://doi.org/10.1007/978-3-031-63778-0_1

[2] Weisfeiler and Leman Go Relational. The First Learning on Graphs Conference 2022. https://openreview.net/forum?id=wY_IYhh6pqj

[3] Neural Bellman-Ford Networks: A General Graph Neural Network Framework for Link Prediction. NeurIPS 2021

[4] Multiplex Graph Representation Learning via Common and Private Information Mining. AAAI 2023

**Questions:**

Please refer to weaknesses section.

---

### Official Review · Reviewer_kFSj · 2024-11-04

**Soundness:** 3
**Presentation:** 3
**Contribution:** 2
**Rating:** 5
**Confidence:** 3

**Summary:**

This paper claims that the current evaluation on multiplex graphs is small-scale and lacks representative features. The evaluation also lacks edge-level tasks, and the performance is saturated. To address these limitations, the authors propose a new benchmark for multiplex graphs, MGB, which includes 10 models, 11 datasets, and 4 tasks. The implementation of the benchmark is publicly available. Findings and insights on existing methods are provided by the authors.

**Strengths:**

- The data, code, and documentation are publicly available, ensuring the reproducibility and accessibility of the benchmark.
- The proposed benchmark is more challenging and offers more opportunities for future research.
- The paper points out possible designs for future models on multiplex graphs.

**Weaknesses:**

- As the authors mentioned, the datasets are still small in scale. The largest dataset only contains 35k nodes.
- Instead of original datasets, the new datasets provided by the authors are processed from existing datasets.

**Questions:**

- Please refrain from indicating the package is released via pip, which is not anonymous, in the readme file during the double-blind reviewing period.
- Adding an example of heterogeneous graphs and multiplex graphs in Section 2.1 can help audiences better understand the difference.
- Line 104, $\mathcal{E}_r$ and $\mathbf{A}_r$ are basically the same information, which means no need to appear twice in the graph definition.

---

### Meta-Review · Area_Chair_8JUK · 2024-12-17

**Metareview:**

This paper proposes a benchmark (MGB) for evaluating graph ML model performance on multiplex graphs.  Authors raise points about weaknesses in existing multiplex graph benchmark setups and propose MGB in order to alleviate these issues, especially the perceived saturation in existing benchmark settings, and run comparisons on this benchmark to showcase observations around modeling tactics for multiplex graphs.

Reviewers leaned towards rejecting the work at this time; moreover, authors did not provide a rebuttal so these assessments did not improve over the review cycle.  There were a few key concerns raised:

- Reviewers are unclear on the positioning and value of this benchmark to the community in the context of other benchmarks, e.g. relational graph benchmark or heterogeneous graph benchmarks (K2hv, pM7H, 27VF)

- There are some unclear and ill-motivated choices in the benchmarking setup (e.g. Pts 2 and 7 raised by K2hv, pM7H)

- The introduced datasets are somewhat small (sub 35k nodes) and mostly preprocessed from existing datasets (27VF, kFSj).  This choice does not appear to be well-justified and seems to inherit some limitations the authors point out about existing datasets.

I encourage the authors to revise their work accordingly.

**Additional Comments On Reviewer Discussion:**

Authors did not provide a rebuttal.

---

### Decision · Program_Chairs · 2025-01-22

Reject